# Biological weed control to relieve millions from *Ambrosia* allergies in Europe

Urs Schaffner [1,12], Sandro Steinbach[2,3,12], Yan Sun[4], Carsten A. Skjøth [5], Letty A. de Weger [6], Suzanne T. Lommen [4,7], Benno A. Augustinus[1,4], Maira Bonini[8], Gerhard Karrer [9], Branko Šikoparija[10], Michel Thibaudon[11] & Heinz Müller-Schärer[4✉]

Invasive alien species (IAS) can substantially affect ecosystem services and human well-being. However, quantitative assessments of their impact on human health are rare and the benefits of implementing IAS management likely to be underestimated. Here we report the effects of the allergenic plant *Ambrosia artemisiifolia* on public health in Europe and the potential impact of the accidentally introduced leaf beetle *Ophraella communa* on the number of patients and healthcare costs. We find that, prior to the establishment of *O. communa*, some 13.5 million persons suffered from *Ambrosia*-induced allergies in Europe, causing costs of Euro 7.4 billion annually. Our projections reveal that biological control of *A. artemisiifolia* will reduce the number of patients by approximately 2.3 million and the health costs by Euro 1.1 billion per year. Our conservative calculations indicate that the currently discussed economic costs of IAS underestimate the real costs and thus also the benefits from biological control.

[1] CABI, 2800 Delémont, Switzerland. [2] Department of Agricultural and Resource Economics, University of Connecticut, Storrs, CT 06269, USA. [3] Department of Management, Technology and Economics, ETH Zurich, 8006 Zurich, Switzerland. [4] Department of Biology, University of Fribourg, 1700 Fribourg, Switzerland. [5] School of Science and the Environment, University of Worcester, Worcester WR2 6AJ, UK. [6] Department of Pulmonology, Leiden University Medical Center, 2300RC Leiden, The Netherlands. [7] Institute of Biology, Leiden University, 2333BE Leiden, The Netherlands. [8] Agency for Health Protection of Metropolitan Area of Milan (ATS), 20122 Milano, Italy. [9] Institute of Botany, University of Natural Resources and Life Sciences, 1180 Vienna, Austria. [10] BioSense Institute - Research Institute for Information Technologies in Biosystems, University of Novi Sad, 21101 Novi Sad, Serbia. [11] French Network of Aerobiological Monitoring RNSA, 69690 Brussieu, France. [12]These authors contributed equally: Urs Schaffner, Sandro Steinbach. ✉email: u.schaffner@cabi.org

As a consequence of globalisation, the number of biological invasions has substantially increased over the past decades, and new introductions do not appear to be slowing down[1]. IAS can have multiple effects on biodiversity and ecosystem services[2] and incur significant economic costs[3–5]. However, while alien parasites, invertebrates and plants are known to cause public health problems[6,7], only a few studies have attempted to quantify their impact on human well-being[4,5,8]. In a recent review of the economic impacts of invasive insects, Bradshaw et al.[4] estimated that the health costs exceed US$6.9 billion per year globally. Yet, studies assesing the impact of invasive species on human health which have been published so far are considered to underestimate the real costs because they are regionally focused, disparate or not grounded in verifiable data[4,5]. This is of concern, as an accurate information of policy and management about the impact of IAS on human health and the potential savings due to the implementation of mitigation measures is essential to ensure that reasonable resources are invested and actions coordinated in IAS management[9].

We apply an interdisciplinary approach to quantify the effects of allergenic pollen produced by common ragweed, *Ambrosia artemisiifolia* L., on human health in Europe. Common ragweed, a native to North America, has invaded different parts of the world[10]. In Europe, it is considered invasive in more than 30 countries[10] and its spread and impact are likely to increase with changing climate[11–13]. Here we show that, prior to the establishment of the biological control agent *Ophraella communa* LeSage in 2013, some 13.5 (95% confidence interval (CI) 10.9–14.8) million persons suffered from *Ambrosia*-induced allergies in Europe, causing economic costs of approximately Euro 7.4 (CI 5.4–8.6) billion annually. Field studies in Italy prove evidence that *O. communa* can reduce *A. artemisiifolia* pollen production by 82%. By modelling the number of generations of *O. communa* across its suitable habitat range in Europe, we project that biological control of *A. artemisiifolia* will, once the leaf beetle has colonised its environmental niche, reduces the number of patients to approximately 11.2 (CI 8.6–12.9) million and the health costs to Euro 6.4 (CI 4.4–7.5) billion per year. Our estimates of the costs of *A. artemisiifolia* for public health are considerably higher than what has been reported previously, suggesting that the actual costs of IAS in Europe and the benefits from their management are underestimated.

## Results

### Effects of *Ambrosia artemisiifolia* on public health.
Based on information from the European pollen monitoring programme, we mapped seasonal total ragweed pollen integrals in Europe during the period 2004–2012, i.e. before the accidental introduction of the North American leaf beetle *O. communa*. We interpolated data from 296 European pollen monitoring sites to a 10 ×10 km grid and extracted them for the European Union (EU) member states and the non-EU member states that are located within the boundaries of the EU (Fig. 1; Supplementary Note 1; Supplementary Figs. 1, 2; Supplementary Data 1).

Pollen allergens produce clinical symptoms only among previously sensitised persons. Therefore, we mapped ragweed sensitisation rates in Europe by combining data on (a) the overall sensitisation rate among the general population, and (b) the ragweed sensitisation rates among the sensitised population (Supplementary Note 2, Supplementary Data 2). The interpolated map of ragweed sensitisation rates in the European population (Fig. 2) corresponds to that of total seasonal ragweed pollen integrals, with both highest pollen integrals and highest ragweed sensitisation rates found in the Pannonian plain, the Po plain and the Rhône valley. By multiplying the interpolated ragweed

sensitisation rates with the European population at a 10 ×10 km grid cell size[14], we estimated that currently 23.2 million persons in Europe are ragweed sensitised (Supplementary Data 3, 4). Taking into account that not all persons with a positive allergy test develop symptoms upon contact with the ragweed allergen, we corrected these numbers for clinical relevance[15], resulting in an estimated 15.8 million persons in Europe with clinically relevant ragweed sensitisation (Supplementary Data 4). Our estimate is 37% lower than that published by Lake et al.[12], which is largely due to our more accurate approach using interpolation of ragweed sensitisation rates based on a large number of geo-referenced locations compared to the region-based approach by Lake et al.[12]. We then determined the population with clinically relevant sensitisation that was exposed to ragweed pollen prior to the establishment of *O. communa* (Fig. 1). This approach resulted in a total of approximately 13.5 (CI 10.9–14.8) million persons in Europe, which likely suffered from seasonal ragweed pollen allergy prior to the arrival of *O. communa* (Supplementary Data 5).

To validate the estimated number of patients suffering from ragweed pollen allergy, we compared our European-wide assessment with detailed healthcare data from the Rhône-Alpes region in southeastern France[16] (www.auvergne-rhone-alpes.ars. sante.fr/; Supplementary Data 6). For each of the region's 313 communities, we calculated the average number of persons which got reimbursed for purchasing anti-allergy or anti-asthma medication during the ragweed flowering season and related them to the average seasonal total ragweed pollen integrals in the communities (the period from 2007 to 2015; Supplementary Figs. 3, 4). We found that seasonal total ragweed pollen counts were significantly correlated with the number of patients, with a 10% decrease in the seasonal total ragweed pollen counts resulting in an 8.4% reduction in the number of patients (Supplementary Fig. 5). With our interpolation approach used at the European level, we estimated that the average proportion of affected patients in the 313 communities in the Rhône-Alpes region was 3.2% (SD 2.2). This figure is similar to the proportion of the population in the communities receiving reimbursements for anti-allergy or anti-asthma medication (2.9%, SD 1.0). Thus, our approach to estimate the number of persons suffering from ragweed pollen allergy appears to be reasonably accurate.

To estimate the European-wide economic costs due to ragweed pollen allergy prior to the establishment of *O. communa*, we estimated European-wide treatment costs per patient and year based on the cost estimates summarised by Bullock et al.[17] for nine European countries. The annual treatment costs varied between Euro 8.30 (for antihistamines in the Czech Republic) and Euro 8,060 (for treatment of asthma in Switzerland), with median treatment costs of Euro 565 per patient and year (Supplementary Note 3). To also account for socio-economic costs, we used the ratio between medical expenses and absence from work calculated for the Rhône-Alpes region (18.5%, Supplementary Data 6). This resulted in estimated annual costs of Euro 670 per patient, a lower and thus a more conservative estimate than the median costs for seasonal allergic rhinitis in Europe (Euro 964 per patient and year[18]). By weighting the treatment and lost work time costs at the country level using purchasing power parity (PPP)-adjusted health expenditures per capita for 2015, we found that the overall economic costs amount to approximately Euro 7.4 (CI 5.4–8.6) billion per year in Europe (Supplementary Note 3; Supplementary Data 7). Our cost estimates are approximately 8-fold higher than those presented by Bullock et al.[17], which is only partly explained by their lower estimate of the medical treatment costs per patient (Euro 303). More importantly, Bullock et al.'s[17] estimates of the number of people affected by ragweed allergy in Europe (between 0.84 and 4.18 million). Our calculations rely on an extensive

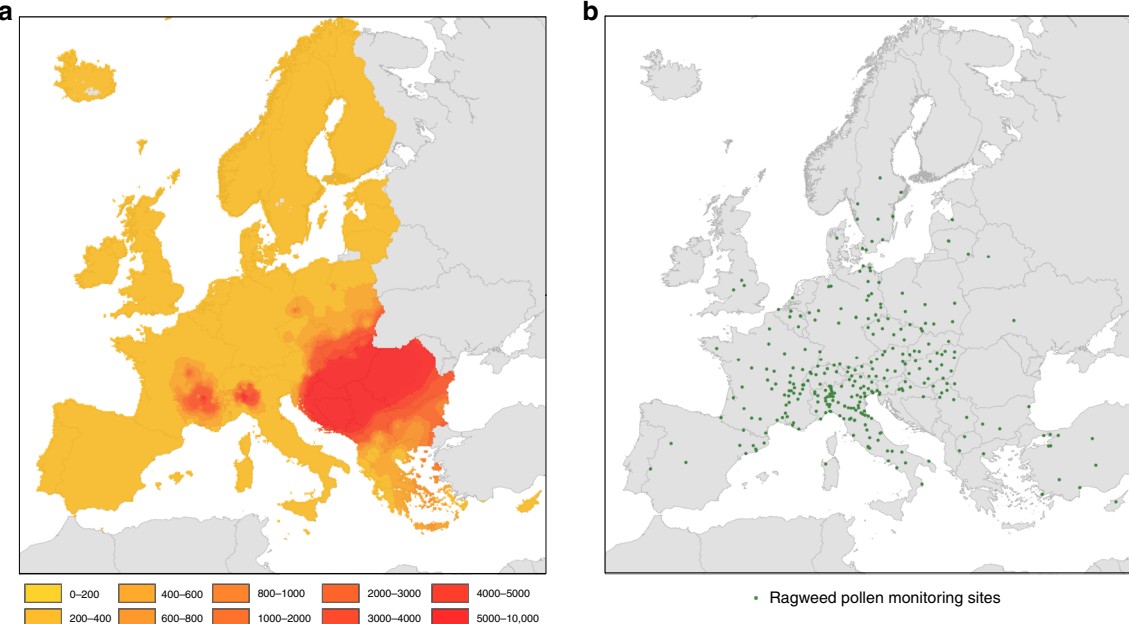

**Fig. 1 Interpolated seasonal pollen integrals for *Ambrosia* across Europe. a** Interpolated seasonal pollen integrals for *Ambrosia* (number of grains per cubic metre of air) across Europe before the establishment of *Ophraella communa* (data from 2004–2012). **b** Seasonal pollen integrals from 296 pollen monitoring stations were used to interpolate ragweed pollen exposure to a 10 × 10 km grid.

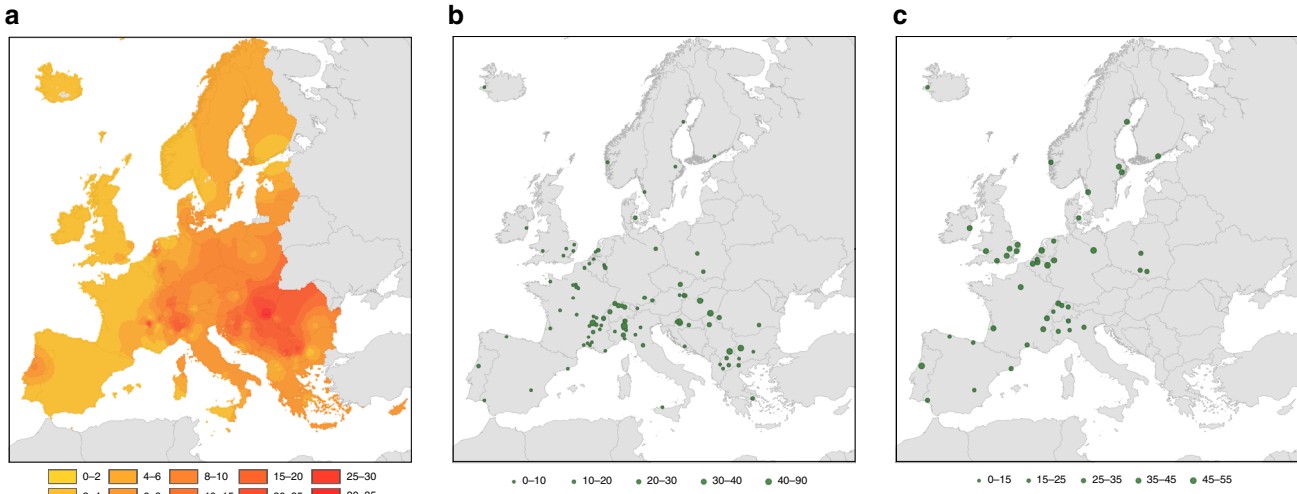

**Fig. 2 Interpolated percentage of ragweed sensitised persons in the European population. a** Interpolated percentage of ragweed sensitised persons in the European population. **b** Geographic distribution of studies assessing overall sensitisation rates among the general population in Europe. Size of points indicates overall sensitisation rates (%) among the persons tested. **c** Geographic distribution of studies assessing ragweed sensitisation rates among the sensitised human population in Europe. Size of points indicates ragweed sensitisation rates (%) among the sensitised persons tested. Studies are based on ragweed specific skin prick tests or circulating ragweed specific IgE tests.

dataset of observed pollen integrals and on a geospatial approach for calculating total number of patients in Europe using a large dataset of observed sensitisation rates. This suggest that our approach is more accurate concerning exposure and that previous numbers of patients have been substantially underestimated. This is supported by the fact that the healthcare data from the Rhône-Alpes region indicate that in this region alone some 200,000 persons per year received reimbursement for purchasing anti-allergy or anti-asthma medication during the ragweed flowering season[16].

**Projected impact of *Ophraella communa*.** The establishment of *O. communa* in Europe in 2013 raised the question whether this

leaf beetle, which is mass-reared and actively distributed in China for biological control of *A. artemisiifolia*[19], might also contribute to the sustainable management of this plant invader in Europe[20]. In Northern Italy, where the beetle was first detected, up to 100% of *A. artemisiifolia* plants are attacked with damage levels high enough to completely prevent flowering of most ragweed plants[20]. Pollen monitoring studies in the Milan area revealed that the substantial drop in airborne ragweed pollen concentrations since the establishment of *O. communa* cannot be explained by meteorological factors[21].

To assess whether the reduction in airborne *Ambrosia* pollen concentrations in Northern Italy indeed reflects the impact of *O. communa* on pollen production of *A. artemisiifolia* plants, we

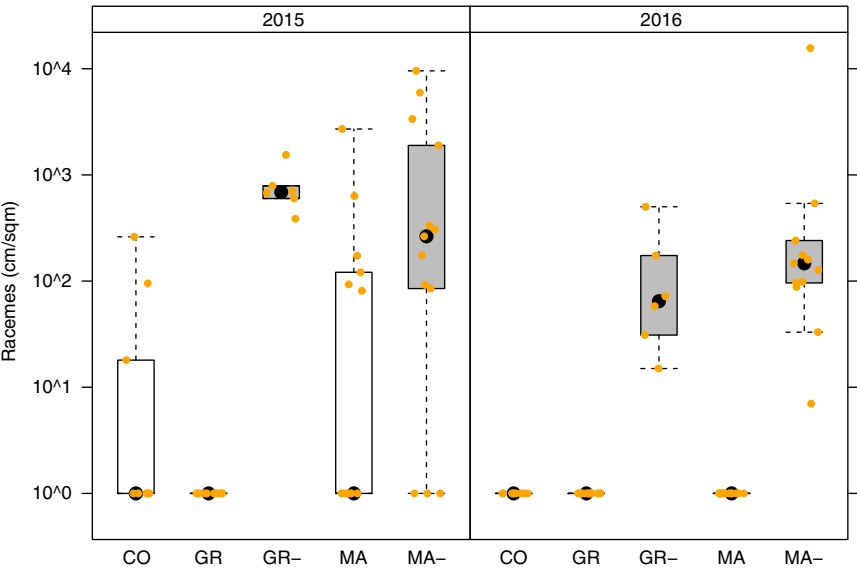

**Fig. 3 Impact of *Ophraella communa* on *Ambrosia artemisiifolia* pollen production.** Estimated *Ambrosia artemisiifolia* raceme density per year at three study sites in Northern Italy in the presence and absence of *Ophraella communa*. CO = Corbetta; GR = Grugliasco; MA = Magnago. GR- and MA- indicate plots from which *O. communa* was excluded by spraying insecticides. At Corbetta permission for insecticide application was not obtained. Boxplots represent the variation across plots within treatment and site, with the black dot as the median, the boxes representing the quartiles, the whiskers 1.5 times the interquartile range and the yellow dots the data points.

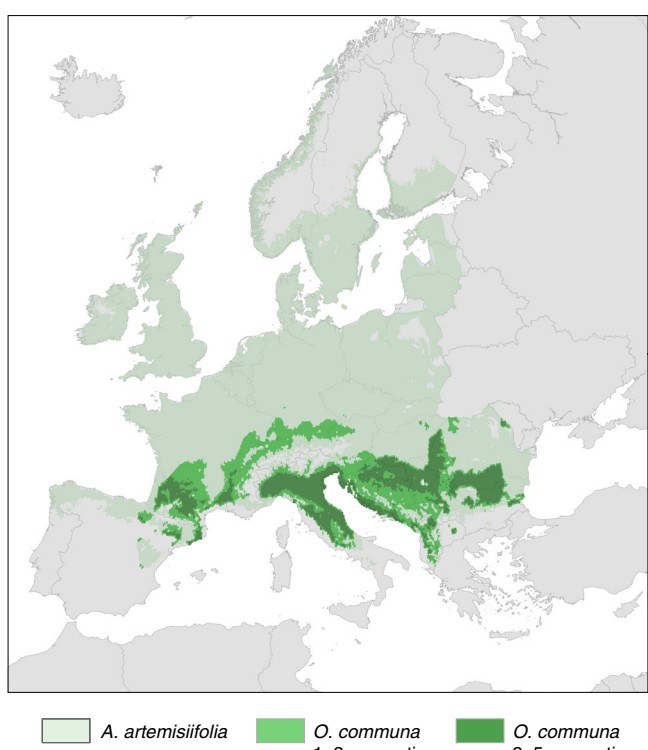

**Fig. 4 Modelled distribution of *Ambrosia artemisiifolia* and *Ophraella communa* in Europe.** The modelled distribution range of *Ophraella communa* includes the expected number of generations per year.

conducted a field experiment in the Po plain during which *O. communa* was excluded in replicated plots at two of three study sites. We found that *O. communa* reduced pollen production on average by 82% (73–100%; Fig. 3), which well corresponds to the drop in airborne pollen concentrations recorded in the Milano area since the establishment of this beetle[21] (Supplementary Fig. 6). Hence, our findings support the notion that the observed

drop in *Ambrosia* pollen integrals in Northern Italy is caused by *O. communa*.

To estimate the potential impact of *O. communa* on ragweed pollen integrals at the European level, we developed species distribution models for *A. artemisiifolia* and *O. communa* based on worldwide occurrences and bioclimatic variables (Fig. 4; Supplementary Fig. 7). We conducted a field cage experiment along an altitudinal gradient in Northern Italy and determined an average growing degree days (GDD) value of 288.7 (Supplementary Data 8) for one generation of *O. communa*. This value was incorporated into the species distribution model to map the number of generations which *O. communa* is likely to complete across its environmental niche in Europe (Fig. 4). We then quantified the potential impact of *O. communa* on ragweed pollen integrals by matching the number of generations of the beetle with the interpolated seasonal pollen integrals (Supplementary Note 1). Based on the association between seasonal total pollen integrals and relative number of patients determined for the Rhône-Alpes region (Supplementary Fig. 5), we projected *O. communa* to reduce the number of patients to approximately 11.2 (CI 8.6–12.9) million once it will have colonised its entire environmental niche in Europe. This corresponds to an average reduction in the number of patients by 2.3 million (16.9%; Supplementary Fig. 8, Supplementary Data 5). Correspondingly, the approximated yearly economic costs are, when accounting for PPP among countries, projected to drop to Euro 6.4 (CI 4.4–7.5) billion. This reduction results in economic savings of approximately Euro 1.1 billion annually (Supplementary Data 7). Besides Northern Italy, where the beetle has already significantly reduced airborne pollen concentrations, our projections suggest that people in countries of the Balkan Peninsula will benefit most from the establishment of *O. communa* (Figs. 2 and 4; Supplementary Data 5).

## Discussion

Our study provides evidence that the impacts of common ragweed on human health and the economy are so far highly underestimated, but that biological control by *O. communa* might mitigate these impacts in parts of Europe. So far, host specificity

studies with closely related crops, ornamentals and native endangered species indicate no significant negative impact of *O. communa* on non-target plants under field conditions[20,22]. The results of our interdisciplinary study justify a comprehensive risk: benefit assessment of *O. communa*, also regarding a possible deliberate distribution of this leaf beetle across the climatically suitable areas in Europe.

Estimates of the economic impact of biological invasions have provided important input in policy and management at the national and international level[3,23]. Our estimate of the costs incurred by *A. artemisiifolia* is in a similar range as the currently discussed overall economic costs of IAS in Europe[24] (Euro 12 billion per year). We propose that future assessments of economic impacts of IAS should more thoroughly consider costs related to human health.

## Methods

**Ragweed pollen exposure map.** The estimates of ragweed pollen exposure in Europe are based on the best available information on long-term exposure to *Ambrosia* pollen for the period from 2004 to 2012. The data were obtained from published work of the mean seasonal total *Ambrosia* pollen integrals from 296 observational sites found in most large urban regions of Europe (Fig. 1b). We incorporated data from all sites where seasonal pollen integrals cover at least five years in order to secure sufficient data points near and outside the main invasion fronts of *A. artemisiifolia* (Supplementary Note 1). The final dataset with 296 calibration points for *Ambrosia* pollen exposure both within and outside the EU was interpolated to a $10 \times 10$ km$^2$ grid using the common European Geographical Reference System (GCS_ETRS_1989). The gridded data were then extracted for the study area which covers 42 countries within Europe and includes EU27 and the bordering non-EU member countries Albania, Serbia, Macedonia, Bosnia-Herzegovina, Kosovo, Montenegro, Iceland, Norway and Switzerland (Supplementary Fig. 1).

**Map of ragweed sensitisation among the European population.** The map of ragweed sensitisation rates among the population in Europe is based on a combination of two types of published studies: (a) overall sensitisation rates among the general population obtained from medical centres and (b) ragweed sensitisation rates among sensitised patients collected from medical centres. A thorough review of all European studies was conducted and all peer-reviewed studies based on skin prick and circulating specific Immunoglobulin E (IgE) testing were included, while questionnaire studies were excluded. The review provided 50 data sets on the sensitisation rate among the general population and 80 data sets on ragweed sensitisation among the sensitised population (Figs. 2b, 2c; Supplementary Note 2; Supplementary Data 2).

The two data sets (50 areas and 102 areas) were then interpolated to a $10 \times 10$ km grid using the common European Geographical reference system (GCS_ETRS_1989). After that, the gridded data were extracted for the study area. Resulting *Ambrosia* sensitisation rates among the European population were then calculated by multiplying the gridded overall sensitisation rates among the population with the ragweed sensitisation rates among sensitised persons. We used natural breaks in the data to classify the exposure and population data (Figs. 1 and 2). Considering that only a proportion of patients with positive skin prick test reactions express symptoms, we calculated the clinically relevant sensitised population by multiplying the ragweed sensitised population with the clinically relevant ragweed sensitisation rates at the country level[15]. When determining the population with clinically relevant sensitisation that was exposed to ragweed pollen prior to the establishment of *O. communa*, we excluded all locations with very low seasonal ragweed pollen estimates (<10 grains per cubic metre during the pollen season[12]).

**Relationship between seasonal *Ambrosia* pollen integrals and patient numbers.** The relationship between the seasonal *Ambrosia* pollen integrals and the number of patients was calculated with information from the database compiled by the 'Agence régionale de santé Auvergne-Rhône-Alpes' (www.auvergne-rhone-alpes.ars.sante.fr/) on health costs related to common ragweed allergies in the Rhône-Alpes region. The Rhône-Alpes region, which is since 1 January 2016 part of the new region Auvergne-Rhône-Alpes, lies in southeastern France and covers an area of 43'700 km$^2$ with approximately 6.5 million inhabitants. The Rhône Valley, which runs north-south through the Rhône-Alpes region, reports the highest common ragweed infestation rates in France[10] and, together with the Pannonian plain and Northern Italy, also the highest rates in Europe[25].

Since 2007, the Regional Health Agency, in association with the National Aerobiological Monitoring Network, has been collecting data on the annual health costs of ragweed allergies in the Rhône-Alpes region[26]. These cost calculations are based on the consumption of medical care and medical goods by persons affiliated to the general health insurance scheme and cover 98.9% of all patients. The included costs relate to allergy medication, doctor consultations, allergy tests, oral desensitisation treatments and sick leave when linked to the prescription of anti-allergy drugs during ragweed flowering season (see Supplementary Data 3 for further explanations). The relationship between total seasonal ragweed pollen integrals and number of patients was calculated for the period 2007–2015 using data from all 313 communities in the Rhône-Alpes region (Supplementary Fig. 5). The number of patients is defined as the number of persons receiving reimbursement through the general health insurance scheme for purchasing anti-allergy or anti-asthma medication to treat allergies during the ragweed pollen season from July 7 to October 21[26].

We interpolated the community-level pollen exposure from pollen data for 62 monitoring sites in France (Supplementary Fig. 3), using the same approach as for the calculation at the European level. Instead of interpolating the annual total pollen integrals at the grid-cell level, we used the community centroid as the central point for interpolation. We found the highest exposure in the area around Lyon and the area on both sides of the river Rhone which is consistent with the habitat preferences of the plant[27] (see Supplementary Fig. 4).

The method for estimating the number of patients is likely to represent the lower bound of the overall effect since only those persons were included which were reimbursed for the consumption of allergy-related medication.

Besides *A. artemisiifolia*, some other plant species flowering in late summer or autumn also possess allergenic pollen, which may theoretically pose some challenges for allergy epidemiological studies. However, in southeastern France, the two other ragweed species (*Ambrosia trifida* L. and *A. psilostachya* DC.) are rare. Moreover, peak flowering of allergenic mugwort species, such as *Artemisia vulgaris* L., *A. absinthium* L. and *A. verlotiorum* Lamotte, is before and after the peak flowering season of common ragweed, and the season of airborne *Artemisia* pollen hardly overlaps with the season of airborne *Ambrosia* pollen[28]. Moreover, seasonal pollen integrals for *Artemisia* pollen are far lower in southeastern France (Supplementary Fig. 4b) and northern Italy (Supplementary Fig. 6d) than those for *A. artemisiifolia* and thus only marginally contribute to the consumption of allergy-related medication in these regions (Supplementary Note 3).

To obtain an estimate of the treatment effect, we averaged the total seasonal pollen integrals and the number of patients for each community and the period 2007–2015. We then log-transformed the data and estimated the relationship without a constant using the linear least squares approach. This approach forces the regression to go through the origin which is a necessary condition for modelling the causal relationship between pollen counts and patient numbers. Overall, the specified regression model fits the data well and explains a significant share of variation (Supplementary Fig. 5).

**Impact of *O. communa* on pollen production.** In 2014, we selected three sites in northern Italy where *A. artemisiifolia* and *O. communa* co-occurred (Corbetta: N45.4709 E8.9368, Magnago: N45.5707 E8.7855, Grugliasco: N45.0654 E7.5923). Half of each site was assigned to experimental exclusion of *O. communa* in an experimental block design, except for the site in Corbetta for which we did not obtain permission to apply insecticides. In each site-treatment combination, 14 permanent plots of $0.5 \times 0.5$ m$^2$ were installed covering the range of *Ambrosia* densities in June 2014 after ragweed had successfully established, and ensuring similar densities across treatments within sites. Starting in July 2014, beetles were experimentally excluded from the blocks assigned by the biweekly application of insecticides while control blocks in these sites were treated with water only. In 2015 and 2016, treatments started in May (corresponding to the period where the first beetle generation of offspring from the overwintering adults develops) and lasted until *O. communa* disappeared from the sites in autumn. The insecticide treatment constituted of spraying three insecticides (including two contact insecticides and a systemic insecticide; for further details see Lommen et al.[29]). This method has no direct effects on seed and pollen output or pollen allergenicity of *A. artemisiifolia*[29].

As a proxy for the yearly pollen production per plot[30], we estimated the total raceme length $r$ per plot in 2015 and 2016 following $r = n * m * e^{v * \beta + \alpha}$, where $n$ is the total number of *A. artemisiifolia* plants in the plot at the end of the season in September (excluding the new seedlings that had just appeared in Magnago in 2015 and which did not produce pollen anymore), $m$ is the fraction of these *A. artemisiifolia* that had matured per site-treatment combination (as a proxy for the fraction of plants that had produced racemes), $v$ is the mean log-transformed volume of the matured *A. artemisiifolia* in the plot (based on all or a sample of ca. 10 plants measured in the plot ($N = 47$ plots), or the site-treatment mean if no plants in the specific plot had been measured ($N = 23$ plots)), and $\alpha$ and $\beta$ are the intercept and the slope, respectively, of the linear relationship between the log-transformed total raceme length and the log-transformed volume of individual matured *A. artemisiifolia*, as assessed by a sample of 21 plants taken outside of the plots per site-treatment combination. The analysis excluded plots that contained no *A. artemisiifolia* plants in 2014 (as these may represent the absence of a soil seed bank), and all plots in one block in Grugliasco where grasses dominated and suppressed ragweed occurrence after 2014. As too few mature plants were available in Corbetta and Magnago in 2016, the respective raceme-volume relationships from 2015 were used instead. The estimated raceme lengths per plot were rescaled to represent racemes per square metre.

Since many plots were estimated to produce a raceme length of 0 (i.e. no racemes), we took a hurdle-model approach for the statistical analysis of the effect of treatment on the raceme length per plot. As in 2016 none of the control plots

and all of the insecticide-treated plots produced racemes, we could only do this for the data from 2015. We first assessed the effect of treatment on the probability of a plot producing racemes in 2015 by a generalised linear mixed effect model (GLMER) with treatment as a fixed effect, site as a random effect, and a binomial error distribution, tested against the corresponding null-model. For those plots producing racemes in 2015, we then similarly assessed the effect of treatment on the total length of racemes produced by a GLMER with treatment as a fixed effect, site as a random effect, and a Poisson error distribution with a log-link, which was tested against the corresponding null-model. The models were then used to obtain fitted estimates of the probability to produce racemes and the total length of racemes produced for each treatment, and their product was an estimate of the density of racemes produced (in cm per m$^2$).

In 2015 the insecticide treatment increased the probability of a plot to produce racemes from 0.26 (95% CI = 0.14–0.42) to 0.84 (95% CI = 0.37-0.98; $\chi_1^2$ = 17.496, $p < 0.001$). When considering plots with raceme production, insecticide-treated plots produced significantly higher densities of racemes than control plots (fitted average 905 cm/m$^2$ versus 260 cm/m$^2$, respectively; $\chi_1^2$ = 6573, $p < 0.001$). As mentioned above, in 2016 only insecticide-treated plots produced racemes (Fig. 3).

**Species distribution models for *A. artemisiifolia* and *O. communa*.** We collected occurrence data for *A. artemisiifolia* and *O. communa* worldwide from the literature, from online resources and from additional occurrence sources (see Appendix S2 in Sun et al.[13]). In summary, we used WORLDCLIM current climate data (developed by Hijmans et al.[31]) at 5-min spatial resolution to derive a set of meaningful predictors that are considered critical to plant or insect physiological function and survival of both species. The final model for *A. artemisiifolia* included annual mean temperature, isothermality, temperature seasonality, max temperature of warmest month, min temperature of coldest month, temperature annual range, mean temperature of driest quarter and an approximation of GDD based on monthly average temperatures (Thom's formula). The final model for *O. communa* included max temperature of warmest month, temperature annual range, mean temperature of driest quarter, mean temperature of coldest quarter, precipitation of wettest month, precipitation of wettest quarter, precipitation of driest quarter, precipitation of coldest quarter and standard insect growing degree days (GDD > 13.3 °C as the minimum development threshold temperature for all immature stages according to Zhou et al.[32])[13]. We modelled the current potential distribution for each species in Europe using the integrative modelling framework Biomod2 R-package[33]. The combination of different modelling techniques generating an ensemble of predictions is proposed as an optimal solution for dealing with the uncertainties of extrapolation[34]. Generalised linear models (GLM), generalised boosting models (GBM), random forest (RF), and maximum entropy (MAXENT) were therefore calibrated on a random sample of the initial data (80%) and tested on the remaining data sets (20%) with both the receiver-operating characteristic curve (ROC) and the true skill statistic (TSS). We then estimated their area under the curve (AUC) that evaluates the discriminatory power of model predictions. Above techniques were chosen because they have proven to presently be among the most effective species distribution models[35]. Duplicated presences within a raster pixel were removed. As only occurrences were available, random pseudo-absences were generated[36] to fill the absence component of models. The entire training-evaluation procedure was repeated 100 times (25 times for each model), using a different set of calibrated presences and absences within each iteration to ensure robustness of the predictions and provide uncertainty estimates[37]. A stacked assemblage of predictions across individual models including mean, inferior confidence interval, superior confidence interval of both *A. artemisiifolia* and *O. communa* were generated. The suitability of species distributions of two species were then binary-transformed using species-specific thresholds maximising the rate of the number of corrected predicted presences to number of false absences (i.e., to transform the probabilities of presence into presence and absence). Based on these thresholds, we also binary-transformed all confidence interval predictions (Supplementary Fig. 7), which were later used for uncertainty estimation. In our models, AUC ranged from 0.87 to 0.96 across the two species and four model types, which provided useful information for an analysis of climate suitability through modelling of the species distributions. Finally, the overlap maps of *O. communa* on its host plant *A. artemisiifolia* that we used for later computations were produced based on their binary mean distribution maps, binary lower confidence interval maps and binary upper confidence interval maps (BIOMOD_EnsembleModeling function in Biomod2)[13].

The number of generations of *O. communa* across the potential distribution in Europe was estimated by experimentally assessing the temperature-dependent developmental time of *O. communa* along an altitudinal gradient in the Southern Alps, approximately 50 km north of the Milan area. The five field sites were set up on sun-exposed grasslands or in private gardens at 130 m (in the Po plain), 250 m, 480 m, 700 m and 1230 m. Greenhouse-reared, potted *A. artemisiifolia* plants were exposed to female *O. communa* in a field cage set up at the lowest elevation. Plants with freshly laid egg batches were individually covered with a gauze bag, firmly attached to the pot with an elastic ribbon, and transferred within 48 h after oviposition to one of the five field sites in a randomised order. At the field site, *A. artemisiifolia* plants were placed within a 1 × 1 × 2 m gauze cage, dug together with the pots into the soil and regularly watered. Two cohorts of 5–8 plants each were transferred to the five field sites, with the exception of the highest site, which only

received plants from the second cohort. The first cohort was set up between 29 June and 2 July 2016 and the second cohort between 8 and 10 August 2016. Each field site was visited at least once a week to record the presence of beetle life stages on each *A. artemisiifolia* plant. Based on the field study, we calculated an average GDD > 13.3°[32] of 288.7 for the period from the egg stage to adult emergence of *O. communa* (Supplementary Data 8). This value closely corresponds to the GDD reported for *O. communa* by Zhou et al.[32] which was based on growth chamber experiments with constant temperature regimes (307.2). We then used the GDD calculated from our field experiment to map the number of generations of *O. communa* in relation to the climatic conditions within its potential distribution range in Europe.

**Impact of *O. communa* on airborne *Ambrosia* pollen concentration in Europe.** To quantify the potential impact of *O. communa* on seasonal pollen integrals in Europe, we exploited retrospective data from northern Italy. We used data on ragweed pollen concentration (number of pollen per cubic metre of air) with a daily resolution for three monitoring sites in the Milan area for the period 2004–2018. The sites are located in proximity (less than 30 km) to the Malpensa airport, where the beetle was initially detected in 2013. Because common ragweed was already established in all potential habitats before the beetle's arrival, we can calculate the pre-treatment exposure and compare it with the post-treatment exposure. For this purpose, we first obtained a measure of the average daily pollen integral for each station during the two periods and then calculated the average of pollen integrals over the three monitoring sites. Supplementary Fig. 6 shows the average impact of *O. communa* on daily pollen integrals in the Milan metropolitan area. The average impact of the beetle on ragweed pollen exposure is 86.3%, which implies that the average pollen integrals dropped from 46.2 to 7.5 pollen grains per cubic metre of air. We used this information to estimate the empirical reduction function of pollen concentration due to *O. communa*. For this purpose, we cumulated the daily pollen integrals before the arrival (2004-2012) and calculated the reduction in pollen concentration at the respective level of pollen concentration in the pre-arrival period. The impact of the beetle is stronger for the lower range of annual pollen concentration (above 90%) and levels out at 83.7% for the higher range (Supplementary Fig. 6b). In terms of human exposure to ragweed pollen, this resulted in a reduction of the number of days with ragweed pollen and reduced pollen integral during days with ragweed pollen (Supplementary Fig. 6c).

In Northern Italy, the population density of *O. communa* and the feeding damage increased significantly in August, which corresponds with the end of the beetle's third generation. We thus assumed that all projected areas with at least three beetle generations face the same reduction in total seasonal ragweed pollen integrals as in the Milan area (86%; Supplementary Figs. 6a and 8). For areas with one or two generations we assumed a reduction in seasonal *Ambrosia* pollen integrals by 30%, due to a likewise reduction in plant densities by early feeding damage that is not compensated for during the growing season (Lommen, unpubl. results). For all other areas, we assumed no effect of *O. communa* on seasonal *Ambrosia* pollen integrals.

**Reporting summary**. Further information on research design is available in the Nature Research Reporting Summary linked to this article.

## Data availability

The source data underlying Fig. 3 and Supplementary Fig. 6 are provided as a Source Data file. All data sets linked to spatial data sets have been deposited in https://eprints.worc.ac.uk/7124/ and can be made available upon request. The data that support the analysis of the percent persons suffering from ragweed allergies in the Rhône-Alpes region are available from the French Network of Aerobiological Monitoring (RNSA) but restrictions apply to the availability of these data, which were used under written agreement for the current study, and so are publicly not available. Data are however available from M. Thibaudon upon reasonable request and with permission of RNSA.

## Code availability

All R, Stata, and ArcGIS codes necessary to produce the results here can be made available upon request.

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

## Acknowledgements

This study was coordinated within the framework of the EU COST Action FA1203 "Sustainable management of *Ambrosia artemisiifolia* across Europe (SMARTER)" (http://www.cost.eu/COST_Actions/fa/FA1203?parties). We acknowledge financial support from the Swiss State Secretariat for Education, Research and Innovation (C14.0063 to US and C13.0146 to HMS), the Swiss National Science Foundation (#31003A_166448 to HMS), the Swiss Federal Office for the Environment (13.0098.KP/M323-0760 to HMS) and that for Agriculture (1062-62,200 to HMS) and the EU COST Action SMARTER. Urs Schaffner was supported by CABI with core financial support from its member countries (see http://www.cabi.org/about-cabi/who-we-work-with/key-donors/). Carsten A. Skjøth was supported by European Commission through a Marie Curie Career Integration Grant (Project ID CIG631745 and Acronym SUPREME), Yan Sun by an Advanced Postdoc Mobility fellowship from the Swiss National Science Foundation (P300PA_161014), Branko Šikoparija was supported by the Ministry of Education, Science and Technological Development of the Republic of Serbia and Suzanne T. Lommen received funding from the European Union's Horizon 2020 research and innovation programme under the Marie Skłodowska-Curie grant agreement No 786624. We thank Teja Tscharntke and Ragan Callaway for commenting on an earlier version of the manuscript.

## Author contributions

The study was conceptualised by U.S., S.S. and H.M.S.; S.S. coordinated data collection and data analysis, with support from C.A.S., L.A.D.W., Y.S., M.B., G.K., B.S., M.T., U.S. and H.M.S.; The field study on the impact of *O. communa* on pollen production was designed, executed and analysed by S.T.L., the species distribution modelling by Y.S., and the demographic study by B.A.A.; U.S. wrote the first draft of the paper, and all authors contributed to writing subsequent versions.

## Competing interests

The authors declare no competing interests.
