## [Peer Review File · Nature Communications]

Reviewers' Comments:

Reviewer #1:

Remarks to the Author:

In this manuscript, the authors conducted an ambitious study on the biological control of a major allergenic plant (*Ambrosia artemisiifolia*) by an accidentally introduced leaf beetle *Ophraella communa* in Europe. Focusing on the sanitary effects of this invasive weed, the authors skilfully combined economic quantifications, field experiments and modelling approaches within an interesting framework. They showed that (i) *Ambrosia*-induced allergies are likely to cost € 9 billion each year in Europe (treatment and socio-economic costs) and (ii) *O. communa* can reduce *A. artemisiifolia* pollen production, and thus the number of patients and the health costs associated. This manuscript provides patterns and figures that are of great interest and value for both public health and management of invasive alien species. I find that this work should merit publication but I would be happy if the authors could first consider the following questions and comments intended to (I hope) improve the Ms.

Overall, the Ms is well written and detailed. However, it is really dense and sometimes not straightforward to follow (it may be just me). I think that adding subheadings to the different sections should improve both clarity and organization of the main text (Lines 42-165). In this case, I guess the authors would have to re-organize their paper by separating intro and discussion. If they much prefer avoiding subheadings, the authors may at least introduce each paragraph by a clear sentence stating the focus/objective(s) of the dedicated section with a clear final statement. To my opinion, this was the case for the paragraph L.124-131 for instance. It would be more impactful if the structure of the main text could simply refer to the different parts of the work (economic quantification including the determination of the number of patients, field studies to characterize relationships between both species and models for projecting potential distribution and impact) although all of them are of course connected, but it is a subjective point of view.

Also, the authors presented a lot of items mostly published as supplementary materials (very nice for full understanding). As the Ms should be self-containing regarding main tables and figures, I wonder whether it would not be useful to replace some items within the main paper (the journal allows max. 10 items if necessary and justified). As the tables are obviously too large, the authors should consider, for instance, to move the supp. figs 2 and 5 in the main Ms as they can help understanding.

While I am afraid that I am unable to provide a robust critical view on some analyses presented (I don't feel expert enough to assess both the relevance and technical validity of interpolation methods), I was really impressed by the large amount of data presented as well as the description of the analyses performed. Nevertheless, I was somewhat concerned (maybe naively) by how the species distribution models performed by the authors were presented in the text. As these models allowed them to draw conclusions on the potential health and financial benefits likely to be obtained from biological control of *A. artemisiifolia* by *O. communa* conclusions, I think that the authors should give more interest to the rationale and implementation of these models, as well as their outcomes.

Especially, unless I missed the info, I did not see any evaluation of the predictive power of the models, which is usually tested using the area under the receiver operator characteristic function (area under the curve criteria, AUC) or the TSS. Also, as SDMs are intrinsically associated to a large variability, forecasted impacts should always be provided with an assessment of their uncertainty (that could also be mapped), so that conclusions as well as management and conservation decisions can be taken in the full knowledge of their reliability. I am convinced that adding at least both elements (model evaluation and uncertainty level) will reinforce the confidence about the results while not necessarily changing the patterns observed. For instance, I wonder to which extent the average growing days (GDD) experimentally determined in Northern Italy can be generalizable to other places in Europe. I imagine that it would be complicated to

gather the same kind of data in different places throughout the continent, but using this value in the models means to assume that it would be the same in all projected areas. Including and showing a level of uncertainty will bring more caution with the results obtained. The same holds for the interpolation of healthcare data taken from the Rhône-Alpes region to the whole of Europe. This is where an indication of uncertainty might be very relevant, as uncertainties are likely to pile up at each step of the methodology.

Reviewer #2:

Remarks to the Author:

The manuscript by Schaffner et al. claims to have quantified the effects of the allergenic plant *Ambrosia artemisiifolia* (ragweed) on public health in Europe and presents an assessment of the impact of the leaf beetle *Ophraella communa* - which was accidentally introduced in Europe and has been used in other geographical areas (e.g. China) for biological control of ragweed - on the number of patients and healthcare expenses.

The topic addressed by the study is of great interest. However, the reviewer has a number of major concerns:

1. Quantification of the effects of ragweed on public health is an overstated claim, not to say totally exaggerated and misleading. First of all, for the estimation and validation of the number of ragweed pollen allergic patients, the authors calculated the average number of persons receiving reimbursement for purchasing symptomatic anti-allergic or anti-asthmatic medication during the ragweed flowering season. The authors overlooked and ignored a number of confounding factors in their simplistic "quantification" approach:

(i) the ragweed flowering season overlaps with the flowering season of other allergenic plants,
(ii) individuals suffering from highly prevalent perennial allergies (e.g. house dust mite, animal dander, indoor moulds) are also purchasing anti-allergic/asthma medication during the flowering season of ragweed,

(iii) the authors did not include in their estimation/quantification allergic patients undergoing immunotherapy (allergy vaccination). Thus, in addition to the number of persons purchasing symptomatic medication, the number of patients receiving immunotherapy should also be taken into consideration.

(iv) in Europe, individuals are not only exposed to ragweed pollen but also to botanically related weeds (e.g. *Artemisia* species). This situation poses some challenges for allergy epidemiological studies, as well as for diagnosis and therapy of weed pollen allergy. It is well documented that the exposure to ragweed and mugwort (*Artemisia vulgaris*) leads to extensive IgE cross-reactivity and clinical symptoms (Asero et al., *Ann Allergy Asthma Immunol* 113:307-313, 2014). Thus, the biological control of ragweed plants might not change the situation for a large proportion of weed-sensitized patients, as pollen originating from related species will trigger allergic symptoms also in the absence of ragweed pollen.

All these variables/confounders were not taken into consideration by the authors.

2. The reviewer's enthusiasm towards the novelty of the findings was considerably dampened by a not cited article "in press" from the same authors (Mouttet et al., *Basic and Applied Ecology*), in which an estimation of the economic benefits of biological control of ragweed by the leaf beetle *Ophraella communa* in southeastern France is described. Basically the same work is now repeated for Europe.

Reviewer #3:

Remarks to the Author:

I have focussed on the costing part... One overall comment applies: the relation between ragweed

and O communa needs bit more clarification. Why, e.g., is there no impact of O communa taking the niche in Cyprus (and some other countries).

Concerning costs some clarifications are needed: median costs are mentioned, however, economic models generally base their estimates on means and not medians. The 19% for sickness absence from Rhone... how representative is it for other areas/countries... the 9 billion cost estimate is not slightly higher than previous ones, but extraordinary more higher...

Were country-specific cost-estimates used (preferred) or one European average? Would a correction be appropriate before country-specific estimates are aggregated? E.g. purchasing power parities...

Reviewers' Comments:

Reviewer #1:

Remarks to the Author:

We have read the new version and the rebuttal to the reviewers, and we are satisfied by both. We think that this study is a strong contribution for applied science and of importance for societies, and that it will be of great interest to the readers of Nature Comm.

Franck Courchamp & Christophe Diagne

Reviewer #2:

Remarks to the Author:

The authors have addressed the reviewer's concerns

Reviewer #3:

Remarks to the Author:

Dears,

thx for the revision.

In the revision, you are maybe just too much blaming things on misunderstanding or misinterpretation, but it is fully your responsibility exactly to avoid that. Please amend text to avoid any confusion with readers.

You seem to not have addressed the question: would a correction be appropriate before country-specific estimates are aggregated? E.g. purchasing power parities...

NCOMMS-19-07132B – Replies to the reviewers' comments:

Reviewer #3 (Remarks to the Author):

In the revision, you are maybe just too much blaming things on misunderstanding or misinterpretation, but it is fully your responsibility exactly to avoid that. Please amend text to avoid any confusion with readers.

Reply: We acknowledge this comment and agree that it is our responsibility to clarify the issues that may have led to misunderstandings. The issues raised by reviewer 3 referred to the data set from the Rhône-Alpes region, which we used for comparing them with the estimates of the number of patients based on our European-wide approach and to estimate the relative effect of a change in seasonal total ragweed pollen integrals on the number of patients.

We started with this in the previous revision by adding the pollen integrals of *Artemisia* in the Rhône-Alpes region (Supplementary Fig. S4B) as well as in Northern Italy (Supplementary Fig. S6D) to show that the contribution of *Artemisia* spp. to the number of persons receiving reimbursement for treatments of pollen-induced allergies in fall was very low (see also L. 268ff of the resubmitted ms). We now added in L.110ff of the Supplementary Information that, despite the fact that exposure to *A. artemisiifolia* and *Artemisia* spp. can lead to extensive IgE cross-reactivity and clinical symptoms (as correctly pointed out by the reviewer), a reduction of the far higher pollen integrals of *A. artemisiifolia* will result in reductions of overall pollen-induced allergies in fall.

We also added to the new paragraph in the Supplementary Note S3 a sentence emphasising that some anti-allergic/asthma medication purchased during the flowering season of ragweed were most likely used to treat other allergies, but that the highly significant relationship shown in Supplementary Fig. S5 reveals that the number of patients in the communities of the Rhône-Alpes region is largely determined by the size of the local *A. artemisiifolia* pollen integral (L.121ff).

Finally, we added in the legend of Table S6 that the row “annual costs for desensitisation” includes the costs for immunotherapies.

We hope that we now provide further evidence that the data from the Rhône-Alpes region are one of the best, if not the best data set on the economic costs of *A. artemisiifolia*-induced allergies worldwide.

You seem to not have addressed the question: would a correction be appropriate before country-specific estimates are aggregated? E.g. purchasing power parities...

Reply: We appreciate this comment by the reviewer since it allowed us to reflect on a different approach to making the cost estimates comparable across Europe. Supplementary Table S7 now presents estimates of medical costs based on the combined treatment and lost work time costs reported in the literature (Euro 670) and compares them with estimates that account for purchasing power parity (PPP). Also, we describe the method used for calculating PPP-adjusted values in the Supplementary Note S3 (L.100ff). We find that these PPP-adjusted estimates for the cost estimates for *Ambrosia*-induced allergies are slightly lower than our initial cost estimates, but they convey a similar picture, i.e. a substantial drop in the number of patients (not affected) and associated medical costs due to *O. communa*, resulting in economic expenditure savings of Euro 1.1 billion annually. Although the PPP approach is debatable (e.g. Taylor and Taylor (2004) "The Purchasing Power Parity

Debate", *Journal of Economic Perspectives*, 18, 135–158), we decided to follow the reviewer's recommendations and adopt the more conservative PPP-adjusted cost estimates in the manuscript.

In the manuscript, we now state in L.125-129 that 'By weighting the treatment and lost work time costs at the country level using purchasing power parity-adjusted health expenditures per capita for 2015, we found that the overall economic costs amount to approximately Euro 7.4 (CI 5.4-8.65.) billion per year in Europe (Supplementary Note S3; Supplementary Table S7).' Also, we now say in L.172-174 that 'Correspondingly, the approximated yearly economic costs are, when accounting for purchasing power parity among countries, projected to drop to Euro 6.4 (CI 4.4-7.5) billion. This reduction results in economic savings of approximately Euro 1.1 billion annually (Supplementary Table S7).'

Reviewers' Comments:

Reviewer #3:

Remarks to the Author:

The authors have adequately addressed my comments

Response to reviewers' comments

Reviewer #3 (Remarks to the Author):

The authors have adequately addressed my comments

Reply: We are happy to hear that we have adequately addressed reviewer 3's comments.

NCOMMS-19-07132A-Z – Replies to the reviewers' comments:

Reviewer #1 (Remarks to the Author):

*In this manuscript, the authors conducted an ambitious study on the biological control of a major allergenic plant (*Ambrosia artemisiifolia*) by an accidentally introduced leaf beetle *Ophraella communa* in Europe. Focusing on the sanitary effects of this invasive weed, the authors skilfully combined economic quantifications, field experiments and modelling approaches within an interesting framework. They showed that (i) *Ambrosia*-induced allergies are likely to cost € 9 billion each year in Europe (treatment and socio-economic costs) and (ii) *O. communa* can reduce *A. artemisiifolia* pollen production, and thus the number of patients and the health costs associated. This manuscript provides patterns and figures that are of great interest and value for both public health and management of invasive alien species.*

I find that this work should merit publication but I would be happy if the authors could first consider the following questions and comments intended to (I hope) improve the Ms.

<

Reply: We appreciate this positive comment.

Overall, the Ms is well written and detailed. However, it is really dense and sometimes not straightforward to follow (it may be just me). I think that adding subheadings to the different sections should improve both clarity and organization of the main text (Lines 42-165). In this case, I guess the authors would have to re-organize their paper by separating intro and discussion. If they much prefer avoiding subheadings, the authors may at least introduce each paragraph by a clear sentence stating the focus/objective(s) of the dedicated section with a clear final statement. To my opinion, this was the case for the paragraph L.124-131 for instance. It would be more impactful if the structure of the main text could simply refer to the different parts of the work (economic quantification including the determination of the number of patients, field studies to characterize relationships between both species and models for projecting potential distribution and impact) although all of them are of course connected, but it is a subjective point of view.

Reply: As we prefer avoiding subheadings, we checked each paragraph and clarified the focus/objective of the section in two instances, i.e. on L. 104 and on L.136 (which refers to the section mentioned by the reviewer). We now feel that the first sentence of each section clearly states the objective of the section and that each section includes a statement of the results.

Also, the authors presented a lot of items mostly published as supplementary materials (very nice for full understanding). As the ms should be self-containing regarding main tables and figures, I wonder whether it would not be useful to replace some items within the main paper (the journal allows max. 10 items if necessary and justified). As the tables are obviously too large, the authors should consider, for instance, to move the supp. figs 2 and 5 in the main Ms as they can help understanding.

Reply: We appreciate this suggestion and have now moved Supp. Figs. S2 (now Fig. 1B) and S3 (now Fig. 2C) to the main ms and added a new Figure (Fig. 2B) showing the geographic reference area of the studies assessing overall sensitisation rates among the

general public. We suggest leaving Fig.S5 in the Supplementary Information as it refers to the data used to validate the main approach taken in this ms.

*While I am afraid that I am unable to provide a robust critical view on some analyses presented (I don't feel expert enough to assess both the relevance and technical validity of interpolation methods), I was really impressed by the large amount of data presented as well as the description of the analyses performed. Nevertheless, I was somewhat concerned (maybe naively) by how the species distribution models performed by the authors were presented in the text. As these models allowed them to draw conclusions on the potential health and financial benefits likely to be obtained from biological control of *A. artemisiifolia* by *O. communa* conclusions, I think that the authors should give more interest to the rationale and implementation of these models, as well as their outcomes. Especially, unless I missed the info, I did not see any evaluation of the predictive power of the models, which is usually tested using the area under the receiver operator characteristic function (area under the curve criteria, AUC) or the TSS. Also, as SDMs are intrinsically associated to a large variability, forecasted impacts should always be provided with an assessment of their uncertainty (that could also be mapped), so that conclusions as well as management and conservation decisions can be taken in the full knowledge of their reliability. I am convinced that adding at least both elements (model evaluation and uncertainty level) will reinforce the confidence about the results while not necessarily changing the patterns observed. For instance, I wonder to which extent the average growing days (GDD) experimentally determined in Northern Italy can be generalizable to other places in Europe. I imagine that it would be complicated to gather the same kind of data in different places throughout the continent, but using this value in the models means to assume that it would be the same in all projected areas. Including and showing a level of uncertainty will bring more caution with the results obtained. The same holds for the interpolation of healthcare data taken from the Rhone-Alpes region to the whole of Europe. This is where an indication of uncertainty might be very relevant, as uncertainties are likely to pile up at each step of the methodology.*

Reply: We would like to thank the reviewer for this important comment. We added information regarding the AUC of the species distribution models for *A. artemisiifolia* and *O. communa* on L.345ff in the Methods section.

We thoroughly discussed the best ways to include the confidence or uncertainty in our analyses and decided to add them in the manuscript as follows:

- 1) we show a map of the standard deviation of the interpolation of the pollen integral in Fig.S2 and considered the upper and lower confidence intervals in the calculation of the confidence interval for the estimated number of patients (Supplementary Table S5) and medical costs (Supplementary Table S7) before and after the arrival of *O. communa* across Europe.
- 2) We also calculated the 95% confidence intervals for the species distribution models of *A. artemisiifolia* and *O. communa* and show the most extreme variation in the relative cover of the modelled distribution of *A. artemisiifolia* by *O. communa* (i.e. the combinations upper *A. artemisiifolia* CI/lower *O. communa* CI and lower *A. artemisiifolia* CI/upper *O. communa* CI) in Supplementary Fig. S7. The uncertainties in the pollen mapping and the species distribution models were then included in the calculations of the confidence intervals for the estimated number of patients and medical costs after establishment of *O. communa*.
- 3) In Europe, the GDD under field conditions can only be assessed in Northern Italy, since this is the only region with an altitudinal cline where the beetle has established

so far. However, our GDD values are similar to those determined by Zhou et al. (2010) under growth chamber conditions, so we are confident that they are generalizable. We now mention this on L.381 in the Methods section.

- 4) As we only used the healthcare data from the Rhône-Alpes region to compare them with the estimates of the number of patients based on our European-wide approach, and to estimate the relative (not the absolute) effect of a change in seasonal total ragweed pollen integrals on the number of patients, we suggest not to include uncertainties in the analysis of the Rhône-Alpes data.

Reviewer #2 (Remarks to the Author):

The topic addressed by the study is of great interest. However, the reviewer has a number of major concerns:

1. Quantification of the effects of ragweed on public health is an overstated claim, not to say totally exaggerated and misleading. First of all, for the estimation and validation of the number of ragweed pollen allergic patients, the authors calculated the average number of persons receiving reimbursement for purchasing symptomatic anti-allergic or anti-asthmatic medication during the ragweed flowering season.

Reply: This comment must be based on a misunderstanding of the approach taken in this paper. As indicated in the main ms (e.g. L. 61-87) and the methods section, our paper is not based on the approach described by Reviewer 2. Rather, we mapped ragweed sensitisation rates in Europe, multiplied the interpolated ragweed sensitisation rates with data for the European population, and then estimated the population with clinically relevant sensitisation that was exposed to ragweed pollen prior to the establishment of *O. communa*.

The authors overlooked and ignored a number of confounding factors in their simplistic “quantification” approach:

(i) the ragweed flowering season overlaps with the flowering season of other allergenic plants,

(ii) individuals suffering from highly prevalent perennial allergies (e.g. house dust mite, animal dander, indoor moulds) are also purchasing anti-allergic/asthma medication during the flowering season of ragweed,

(iii) the authors did not include in their estimation/quantification allergic patients undergoing immunotherapy (allergy vaccination). Thus, in addition to the number of persons purchasing symptomatic medication, the number of patients receiving immunotherapy should also be taken into consideration.

*(iv) in Europe, individuals are not only exposed to ragweed pollen but also to botanically related weeds (e.g. *Artemisia* species). This situation poses some challenges for allergy epidemiological studies, as well as for diagnosis and therapy of weed pollen allergy. It is well documented that the exposure to ragweed and mugwort (*Artemisia vulgaris*) leads to extensive IgE cross-reactivity and clinical symptoms (Asero et al., *Ann Allergy Asthma Immunol* 113:307-313, 2014). Thus, the biological control of ragweed plants might not change the situation for a large proportion of weed-sensitized patients, as pollen originating from related species will trigger allergic symptoms also in the absence of ragweed pollen. All these variables/confounders were not taken into consideration by the authors.*

Reply: As indicated in the previous reply, most of the comments appear to be based on a misunderstanding of the overall approach taken in the ms. However, they are partly relevant for the validation study (the data from Rhône-Alpes), where the number of patients was estimated as the number of persons receiving reimbursement for purchasing symptomatic anti-allergy or anti-asthma medication during the ragweed flowering season. Our replies below therefore only refer to the section of the ms, which uses the data from the Rhône-Alpes region to validate our overall approach taken in this ms.

Regarding (i): As mentioned in the main manuscript (L.255ff), there is little overlap of the peak flowering season of *A. artemisiifolia* and other allergenic plants, including *Artemisia* spp. in southern Europe, i.e. in the areas where *O. communa* is expected to reduce pollen concentrations. Even more importantly, the integrals of *Artemisia* pollen in the Rhône-Alpes region (Supplementary Fig. S4B) and in the Milano area (Supplementary Fig.S6D) are far lower than those of *Ambrosia* pollen. Hence, relative to *Ambrosia*, the contribution of *Artemisia* to the number of persons receiving reimbursement for treatments of pollen-induced allergies in fall is very low. We now state this in the ms on L. 258ff.

Regarding (ii) and (iii): Reviewer 2 correctly states that some persons may also purchase anti-allergy or anti-asthma medication for other reasons during the flowering period of ragweed. However, to reiterate the point made above, all these comments are not of direct relevance to this manuscript, since the general approach chosen to estimate ragweed patients (exposure of clinically relevant ragweed sensitised persons to ragweed pollen) and the associated costs (studies on ragweed-specific costs summarized in Bullock et al.) is not based on data from the Rhône-Alpes region.

2. The reviewer's enthusiasm towards the novelty of the findings was considerably dampened by a not cited article "in press" from the same authors (Moultet et al., Basic and Applied Ecology), in which an estimation of the economic benefits of biological control of ragweed by the leaf beetle Ophraella communa in southeastern France is described. Basically the same work is now repeated for Europe.

Reply: We did not cite the article by Moultet et al. (which has now been published in Basic and Applied Ecology), because a different approach was used in our manuscript. Moultet et al indeed used the approach described by Reviewer 2. However, the approach taken by Moultet et al. does not allow upscaling of number of patients and costs, nor does it allow to assess the effect of *O. communa* on airborne pollen concentrations. We, therefore, would like to emphasize that there is no repetition of work between the two studies. Nevertheless, we now cite the study by Moultet et al. conducted in the Rhône-Alpes region (L.90 and L.127).

Reviewer #4 (Remarks to the Author):

I have focussed on the costing part... One overall comment applies: the relation between ragweed and O communa needs bit more clarification. Why, e.g., is there no impact of O communa taking the niche in Cyprus (and some other countries).

Reply: As can be seen in Fig.4, *O. communa* cannot cause impacts in Cyprus because the host plant does presently not occur there and is also not predicted to occur there.

Concerning costs some clarifications are needed: median costs are mentioned, however, economic models generally base their estimates on means and not medians.

Reply: We agree with the reviewer's comment. However; as we now emphasize in the Supplementary Note 3, the distribution of the data set summarized by Bullock et al. is skewed and considerably affected by one very large value for Switzerland (high-medical cost country which is not representative of most of the other countries). We therefore prefer to stick to the (lower and thus more conservative estimate) median costs, both when reporting the economic costs for ragweed pollen allergies as well as to those for seasonal allergic rhinitis in Europe.

The 19% for sickness absence from Rhone... how representative is it for other areas/countries... the 9 billion cost estimate is not slightly higher than previous ones, but extraordinary more higher...

Reply: Just to be sure that there is no confusion: we did not state that the sickness absence in Rhone is 19%. Instead we state "...the ratio between medical expenses and absenteeism calculated for the Rhône-Alpes region (18.5%, Supplementary Note 3, Supplementary Table S6). In Supplementary Table S6 the ratios between absenteeism and medical costs are shown for various years, i.e on average 18.5%. This ratio is not particularly high when compared to other studies discussed in a recent review on allergic rhinitis (Wise, S. K., et al. 2018. 'International Consensus Statement on Allergy and Rhinology: Allergic Rhinitis', Int Forum Allergy Rhinol, 8: 108-352).

We now mention in the Supplementary Note 3 that in their European review Bullock *et al.* estimated that the costs for sickness absence relative to medical costs are 25%. We therefore think that the percentage used in our study is representative for other parts of Europe. Furthermore, the estimated annual costs of Euro 670 per patient, used as basis for the calculation of the Euro 9 billion estimate, is a rather conservative estimate (see also lines 111-115)

Were country-specific cost-estimates used (preferred) or one European average? Would a correction be appropriate before country-specific estimates are aggregated? E.g. purchasing power parities...

Reply: While we agree that country-specific costs would be preferable, there are only between one and four data available from nine countries and none from any of the other countries included in our study. We therefore prefer to provide one European average value. To address the concern raised by the reviewer, we now state in the Supplementary Note 3 that due to a lack of country-specific data we decided to estimate a European-wide value.